# Effect of the time to antivenom administration on recovery from snakebite envenoming-related coagulopathy in French Guiana

Stéphanie Houcke[1], Jean Marc Pujo[2], Segolene Vauquelin[1], Guy Roger Lontsi Ngoula[1], Severine Matheus[1], Flaubert NkontCho[3], Magalie Pierre-Demar[4,5], José María Gutiérrez[6], Dabor Resiere[7], Didier Hommel[1], Hatem Kallel[5]*

1 Intensive Care Unit, Cayenne General Hospital, Cayenne, French Guiana, France, 2 Emergency department, Cayenne General Hospital, Cayenne, French Guiana, France, 3 Pharmacy department, Cayenne General Hospital, Cayenne, French Guiana, France, 4 Laboratory department, Cayenne General Hospital, Cayenne, French Guiana, France, 5 Tropical Biome and immunopathology CNRS UMR-9017, Inserm U 1019, Université de Guyane, Cayenne, French Guiana, France, 6 Instituto Clodomiro Picado, Facultad de Microbiología, Universidad de Costa Rica, San José, Costa Rica, 7 Intensive Care Unit, Martinique University Hospital, Martinique, France

* Kallelhat@yahoo.fr

**Data Availability Statement:** All data used to draw the conclusions outlined in our manuscript are

## Abstract

### Background

Snakebite (SB) envenoming is an acute emergency requiring an early care delivery. We aimed to search for the time to reach healthcare facilities in various regions of French Guiana (FG) and to assess the impact of time to antivenom (AV) on the correction of coagulation parameters in these patients.

### Methodology

This is a prospective observational study conducted in Cayenne General Hospital between January 1st, 2016, and July 31st, 2022. We included all patients hospitalized for SB envenoming less than 48h after the bite, and receiving antivenom (AV). We assessed the time lapse between SB and medical attention and the time needed to return of the coagulation parameters to normal.

### Principal findings

Overall, 119 patients were investigated, and 48.7% were from remote areas. The median time from SB to AV therapy was 09:15 h (05:32–17:47). The time was longer in patients from remote rural locations. AV was dispensed within the first six hours after the SB in 45 cases (37.8%). Time from SB to reaching normal plasma fibrinogen concentration was 23:27 h (20:00–27:10) in patients receiving AV≤6h vs. 31:23 h (24:00–45:05) in those receiving AV>6h (p<0.001). Whereas, the time from AV administration to reach normal fibrinogen dosage was similar in the two groups.

available and freely accessible from the Supporting Information file (S1 Data).

**Funding:** The authors received no specific funding for this work.

**Competing interests:** The authors have declared that no competing interests exist.

## Conclusions

Patients from rural settings in FG suffer from a delay in AV administration after SB envenoming leading to an extended time in which patients are coagulopathic. Once AV is administered, clotting parameters recover at a similar rate. Supplying remote healthcare facilities with AV and with medical teams trained on its use should be planned.

## Author summary

Snakebite envenoming is a public health concern in the Amazon region. It represents an acute medical emergency needing early care such as stroke, severe trauma, myocardial infarction, etc. Antivenoms are the most effective treatment of snakebite envenomings. They are part of the *WHO List of essential medicines* and should be available in any primary health care where snakebite victims are managed. In this context, less than 6 hours delay between the snakebite and the antivenom administration is needed for a maximal chance to prevent and reverse most of the toxic effects of the snake venom.

In French Guiana, like in the Amazon region, antivenoms are available in hospital settings but not in remote healthcare facilities. This leads to extended delays in antivenom administration and, consequently, lesser chances for envenomed victims.

Health authorities should promote supplying primary health care facilities with antivenoms for the optimal management of snakebite victims and an increased chance to reverse the envenoming signs. Moreover, time to antivenom should be considered as an indicator of equality and equity of access to health care for people living in remote and rural communities.

## Introduction

Snakebite (SB) envenoming is a public health concern in the Amazon region [1]. It is a frequent event in French Guiana (FG), where, on average, 90 snakebite envenomings are recorded yearly [1–3]. Typically, the first signs of envenoming appear within half an hour after the bite, and early treatment by antivenom (AV) is recommended to prevent serious complications [3]. In 2017, the French Health Authority decided the use of AV in FG. The AV chosen was the Antivipmyn Tri, deemed effective against most of the snakes encountered in the region [4]. However, the efficacy of this AV remains controversial [2,5,6], and one of the causes of poor efficacy in some cases might be a delayed AV administration. It is noteworthy that AV is available only in the three territorial hospitals, all located along the coastline of FG. While there are 18 remote health centers scattered along the borders with Brazil and Suriname, none of them is provided with AV. Meanwhile, most of the time, the rapid access to the emergency department depends on air means managed by the Regional Call Center for Emergencies based in Cayenne.

The WHO defined SB envenoming as an acute emergency requiring an early care delivery [7]. Indeed, early AV use (within the first 6h of envenoming) in South America and Martinique has been reported to be associated with a better outcome [8–11]. In the Brazilian Amazon, a study of 9,191 SB cases found that a delay of six or more hours in medical care was associated with increased severity of envenoming [12]. However, in FG, twenty percent of the whole population (estimated in 2018 at 296,700 inhabitants in the legal situation) lives in

remote areas in the middle of tropical rainforests that are only accessible by boat or plane. Consequently, patients can take several hours before hospital admission and AV administration. In two recent studies in FG the average time from SB to AV administration was 11:00 h (6:00–20:00) and 09:00 h (05:22–20:40), respectively [2,6].

We conducted this study to search for the time lapse for reaching medical attention in various settings in FG, and to assess the impact of time to AV administration after the bite on the recovery of SB related coagulopathy and other clinical and laboratory parameters.

## Methods

### Ethics statement

Our study is an observational, non-interventional work. The antivenom used is authorized "compassionally" by the French Agency for Drug Safety (code product: 3400893189627). The hospital's institutional review board (Direction qualité du Centre Hospitalier de Cayenne) and the Cayenne Hospital ethics committee (Comité d'éthique du Centre Hospitalier de Cayenne) approved the protocol of antivenom administration and blood test dosages (Ref: UF3700/17', version "b"). We informed all patients about the hospital protocol on the management of SB and that the data collected would be used in research programs. Formal verbal consent was obtained from all patients or parent/guardian (when patients are <18 years or unable to consent) and was reported in the patient's medical file. In addition, at admission to our hospital, an information booklet was distributed to all patients or their relatives stating that their data can be used for research purposes and that they can oppose to this use. The database has been registered at the Commission Nationale de l'Informatique et des Libertés (registration n˚ 2217025v0), in compliance with French law on electronic data sources.

### Study design

Our study is prospective and observational. It was conducted in Cayenne General Hospital between January 1st, 2016, and July 31st, 2022. We included all patients hospitalized with clinical and biological signs of SB envenoming lasting less than 48 hours after the bite, regardless of the grade of envenoming, and receiving antivenom. We excluded all patients hospitalized for SB envenoming lasting more than 48 hours after the bite, those who did not receive antivenom, and those who did not present clinical or biological signs of envenoming.

The study design and the regional AV protocol were previously described [2]. The AV used is Antivipmyn Tri, manufactured and marketed since 2008 by Instituto Bioclon, Mexico. Antivipmyn Tri is a freeze-dried F(ab')$_2$ polyvalent antivenom produced by Instituto Bioclon, in Mexico-Mexico Registry N 58583 SSA IV. It is prepared by immunizing horses with venoms of *Bothrops asper*, *Crotalus durissus terrificus*, *and Lachesis muta*. According to the manufacturer, it is indicated for the treatment of envenoming by vipers, such as *Bothrops atrox*, *Bothrops brazili*, *Bothrops asper*, *Bothrops neuwiedii*, *Bothrops alternatus*, *Bothrops jararacussu*, *Bothrops venezuelensis*, *Bothrops pictus*, *Crotalus durissus terrificus*, *Crotalus durissus durissus*, *Lachesis stenophrys*, *Lachesis muta muta*, *Sistrurus* spp., and *Agkistrodon* spp. The protocol of Antivipmyn Tri used was already described (6 vials whatever the grade of envenoming), and its efficacy was already studied [2,13].

In all envenomed patients, we collected epidemiological and clinical data, including the geographic zone where the bite occurred, age and gender of patients, the date and time of the bite, the anatomical site of the bite, the snake identification, the grade of severity of envenoming, the clinical manifestations and laboratory tests at admission and during the hospital stay, the date and time of hospital admission and AV administration, and the adverse reactions to AV.

## Definitions

The culprit snake was identified based on the patient description, photographs, or on the physical examination of the captured snake. The grade of envenoming was assessed according to the conclusions of the international symposium held in French Guiana in 2017 [3]. It was evaluated at patient admission to the medical service and at hospital discharge (referring to the case evaluation along the whole evolution). The grading system is based on three grades (I: mild, II: moderate, and III: severe) including a large number of signs and symptoms (Table 1).

Worsening skin lesions refer to expanding local edema, necrosis, or blisters over 24 hours of surveillance. Expanding cutaneous edema refers to enlargement of the edema zone by more than 5 cm in size. Expanding necrosis refers to enlargement of the necrosis zone by more than 5 cm in size. Expanding blisters refers to the development of new blisters. Expanding skin manifestation is tracked by drawing a line around the lesion area with a marker and checking whether the lesion extends past the line after 24 hours of medical observation. Acute kidney injury is characterized according to the definition of Kidney Disease Improving Global Outcomes (KDIGO) definition [14]. Fibrinogen was measured by chronometric determination (Closs method) with a threshold of detection of 0.35 g/L. Detectable fibrinogen is defined as a fibrinogen concentration higher than the threshold of detection of the dosage technique. Defibrinogenation is defined by a fibrinogen level <1 g/L (normal range: 2–4 g/L). Thrombocytopenia is defined by a platelet count < 150 G/L. Rhabdomyolysis is defined by a serum creatine kinase (CK) activity > 500 IU/L (normal range: 39–308 IU/L). Hemolysis is defined by an increased serum lactate dehydrogenase (normal range: 105–333 IU/L), increased unconjugated bilirubin (normal range: 0.2–0.8 mg/dL), and decreased haptoglobin levels (normal range: 41–165 mg/dL) in serum [15]. Presence of schistocytes is considered significant when detected at more than 1% on the blood smear test. Coagulation disorders are defined by International Normalized Ratio (INR) >2 (normal range: 0.8–1.2), activated partial thromboplastin time (aPTT) > 1.5, Prothrombin time and concentration of coagulation factors < 60%. Coagulation factors are dosed only when the prothrombin time is less than 60%. According to the hospital protocol, laboratory tests were performed every 6 hours from hospitalization until recovery of the coagulation disorders (fibrinogen >1.5 g/L).

The time to treatment (the interval between the SB and the initiation of medical care) was classified as early (≤6h) or delayed (>6h). Adverse reactions to AV were classified as 'mild' or 'severe' [16,17]. Worsening of clinical manifestations is defined as a progression to a higher grade of envenoming. Patients are hospitalized in ICU until recovery of the coagulation

**Table 1. The grading system for classifying the severity of envenoming.**

|  |  | Grade | | |
|  |  | **I** | **II** | **III** |
|---|---|---|---|---|
| Coagulation disorder* |  | + | + | + |
| Local signs | Pain | + | + | + |
|  | Swelling | Not exceeding elbow or knee | Exceeding elbow or knee | Beyond the root of the limb |
|  | Blister | - | + | + |
|  | Necrosis | - | - | + |
| Local bleeding |  | - | + | + |
| Systemic bleeding |  | - | - | + |
| Systemic manifestations (Hypotension, Renal failure, Coma, Respiratory failure) |  | - | - | Organ failure |

*Based on results of the 20-min whole blood clotting test and laboratory analysis.

disorders. Then, they are transferred to the surgical ward and followed until day 8 from the bite or until recovery of the skin lesion, when it takes more time to heal. After that, patients are followed in the outpatient clinic one week after the hospital discharge.

## Data analysis

We created a data file with the patient's and snake's information, and we performed a descriptive analysis using Excel (2007) and IBM SPSS Statistics for Windows, version 24 (IBM Corp., Armonk, NY, USA). Results are reported as the median and inter-quartile range (IQR), mean and standard deviation, or numbers with percentages. Time is expressed as hours and minutes (hh: mn). To compare qualitative variables, we used the Fisher exact test. To compare quantitative variables, we used the Mann–Whitney U-test. The significance level was set at $p \leq 0.05$. We used Kaplan-Meier analysis to compare the time needed to achieve normal fibrinogen dosage between the early and the late AV groups. The significance level was set at Log-Rank $\leq 0.05$.

## Results

During the study period, 198 patients were admitted to the emergency department of Cayenne General Hospital with a diagnosis of SB envenoming (on average, 35 cases per year). Of them, 111 patients (56%) were transferred from remote rural healthcare centers.

Fig 1 shows the primary sites of consultation, the time from SB to the hospital (one-trip access time by helicopter), and the median and interquartile range of time from the SB to AV therapy according to the geographic region where the SB occurred. Among patients admitted for SB, 119 (60%) reached the inclusion criteria and were analyzed in our study (Fig 2).

The median age of included patients was 41 years (IQR: 27–53), and 67.2% were male. The median time from the SB to hospitalization was 6:13 h (IQR: 1:33–18:10). The median time from the SB to AV therapy was 9:15 h (IQR: 5:32–17:47). The responsible snake was identified in 52 cases (43.7%). It was *Bothrops atrox* in 51 cases and *B. bilineatus* in one case. A higher proportion of patients first attended in remote rural health centers had a delayed (>6h) administration of antivenom as compared to those first attended in urban centers (Table 1).

The main clinical signs and symptoms at admission were edema (97.5%), pain (99.2%), blister (10.1%), local hemorrhage (12.6%), acute kidney injury (17.6%), and systemic bleeding (10.9%). The only parameter showing a statistical difference between the two groups (early and delayed AV) was the local hemorrhage (24.4 *vs*. 5.4%, p = 0.002). The elapsed time from SB to the development of systemic bleeding was 1h (0–4) as per information provided by the patients. During evolution, infection was recorded in 26.1% of cases. Clinical progression of symptoms was recorded in 30 patients (25.2%). It was responsible for envenoming-grade progression in 12/119 cases (10.1%). Table 2 summarizes the epidemiological and clinical parameters of patients.

Antivenom was administered within the six first hours in 45 patients (37.8%) and was delayed more than six hours in 74 patients (62.2%). Early AV administration concerned 23/60 (38.3%), 15/31 (48.4%), and 7/28 (25%) in grade I, II, and III patients, respectively. It concerned 33/61 (54.1%) and 12/58 (20.7%) patients coming from the coastline and consulting at the emergency department, and patients from the remote rural zones and consulting at the remote healthcare centers, respectively. Early adverse reaction during AV administration was observed in 15 patients (12.6%). It was mild in 11 patients (11%) and severe in 4 patients (3.4%). Symptomatic management was based on analgesics (100%), fluid infusion (59.7%), blood components transfusion (4.2%), and dialysis (2.5%). Surgery was required for 27 patients (22.7%), and necrosectomy was performed on 12 (44.4%) of them. The delay from

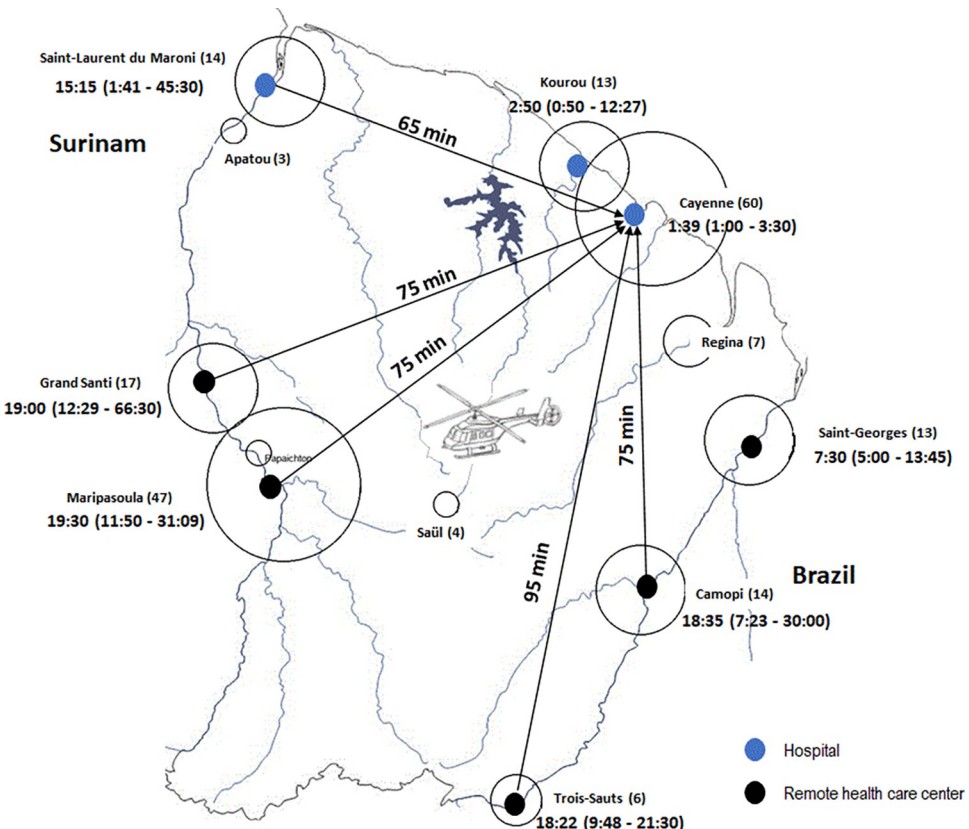

**Fig 1. Place of the initial medical care (black and blue dots), number of cases registered at each place (in parentheses), the one-way time to medical evacuation of envenomed patients to Cayenne hospital by helicopter (minutes indicated in the arrows), and the time to receive AV (hours and minutes, with the corresponding quartiles in parentheses).** The base layer of the map was drawn by hand from the U.S. Geological Survey (http://www.usgs.gov).

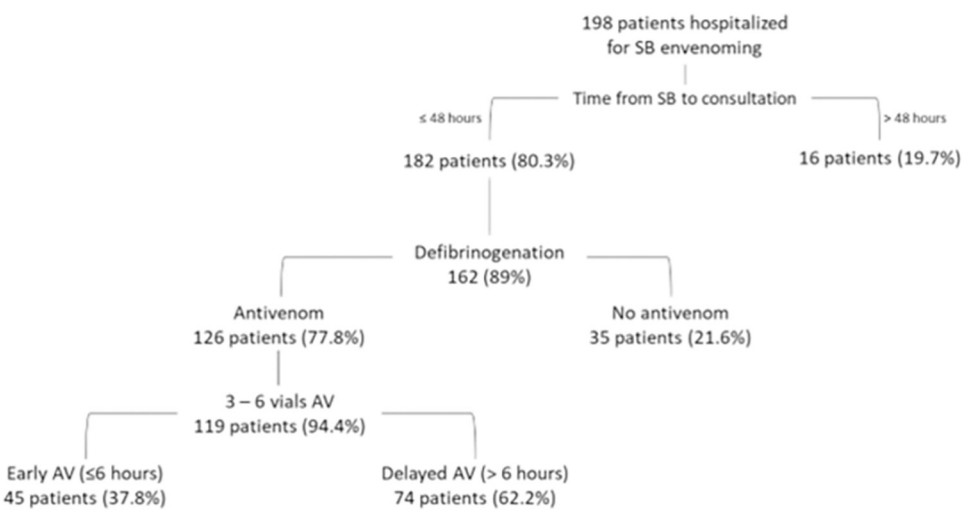

**Fig 2. The flow-chart of the study.**

**Table 2. Epidemiological and clinical parameters at admission.**

| Parameter | Total Nb | Total Result | Early AV Nb | Early AV Result | Delayed AV Nb | Delayed AV Result | p |
|---|---|---|---|---|---|---|---|
| Age, years | 119 | 41 (27–53) | 45 | 38 (25–57) | 74 | 42 (30–53) | 0.530 |
| Male gender | 119 | 80 (67.2%) | 45 | 29 (64.4%) | 74 | 51 (68.9%) | 0.614 |
| BMI, Kg/m$^2$ | 73 | 23.8 (21.2–26.3) | 32 | 23.8 (21.4–26.1) | 41 | 23.8 (21.2–26.4) | 0.960 |
| Past medical history | 119 | 30 (25.2%) | 45 | 14 (31.1%) | 74 | 16 (21.6%) | 0.248 |
| Arterial hypertension | 119 | 10 (8.4%) | 45 | 6 (13.3%) | 74 | 4 (5.4%) | 0.131 |
| Alcohol abuse | 119 | 4 (3.4%) | 45 | 1 (2.2%) | 74 | 3 (4.1%) | 0.591 |
| First attended in a remote health center | 119 | 58 (48.7%) | 45 | 12 (26.7%) | 74 | 46 (62.2%) | 0.000 |
| Snake identification | 119 | 52 (43.7%) | 45 | 28 (62.2%) | 74 | 24 (32.4%) | 0.001 |
| Time from SB to hospitalization | 119 | 06:13 (01:33–18:10) | 45 | 01:30 (00:45–03:00) | 74 | 09:44 (06:03–21:36) | 0.000 |
| **Grade of envenoming on admission** | | | | | | | |
| Grade I | 119 | 60 (50.4%) | 45 | 23 (51.1%) | 74 | 37 (50%) | 0.219* |
| Grade II | 119 | 31 (26.1%) | 45 | 15 (33.3%) | 74 | 16 (21.6%) | 0.064$ |
| Grade III | 119 | 28 (23.5%) | 45 | 7 (15.6%) | 74 | 21 (28.4%) | 0.110£ |
| **Progression of the grade of envenoming#** | **119** | **12 (10.1%)** | **45** | **4 (8.9%)** | **74** | **8 (10.8%)** | **0.736** |
| From grade I to grade II | 58 | 7 (12.1%) | 23 | 3 (13%) | 35 | 4 (11.4%) | 0.853 |
| From grade I to grade III | 53 | 2 (3.8%) | 20 | 0 (0%) | 33 | 2 (6.1%) | 0.521 |
| From grade II to grade III | 31 | 3 (9.7%) | 15 | 1 (6.7%) | 16 | 2 (12.5%) | 0.583 |
| Symptoms related to grade progression | | | | | | | |
| Expanding local edema | 12 | 10 (83.3%) | 4 | 3 (75%) | 8 | 7 (87.5%) | 0.584 |
| Expanding blisters | 12 | 3 (25%) | 4 | 2 (50%) | 8 | 1 (12.5%) | 0.157 |
| **Clinical parameters during hospitalization** | | | | | | | |
| Local edema | 119 | 116 (97.5%) | 45 | 45 (100%) | 74 | 71 (95.9%) | 0.171 |
| Local hemorrhage | 119 | 15 (12.6%) | 45 | 11 (24.4%) | 74 | 4 (5.4%) | 0.002 |
| Necrosis | 119 | 14 (11.8%) | 45 | 3 (6.7%) | 74 | 11 (14.9%) | 0.178 |
| Blisters | 119 | 12 (10.1%) | 45 | 5 (11.1%) | 74 | 7 (9.5%) | 0.772 |
| Pain | 119 | 118 (99.2%) | 45 | 45 (100%) | 74 | 73 (98.6%) | 0.434 |
| Worsening skin lesions# | 119 | 30 (25.2%) | 45 | 11 (24.4%) | 74 | 19 (25.7%) | 0.881 |
| Expanding local edema | 119 | 27 (22.7%) | 45 | 10 (24.4%) | 74 | 17 (21.6%) | 0.924 |
| Expanding blisters | 119 | 5 (4.2%) | 45 | 2 (4.4%) | 74 | 3 (4.1%) | 1.000 |
| Expanding necrosis | 119 | 3 (2.5%) | 45 | 0 (0%) | 74 | 3 (4.1%) | 0.289 |
| Acute kidney injury at admission | 119 | 21 (17.6%) | 45 | 6 (13.3%) | 74 | 15 (20.3%) | 0.336 |
| Time from renal injury to normal renal parameters, days | 21 | 5 (2–10) | 6 | 9 (3–10) | 15 | 3 (2–8) | 0.288 |
| Systemic bleeding | 119 | 13 (10.9%) | 45 | 4 (8.9%) | 74 | 9 (12.2%) | 0.579 |

Nb: the number of cases in whom the parameter was analyzed, Values are expressed as number and percentages or median and interquartile range, BMI: Body Mass Index$: grade II vs. III

£: grade III vs. I and II *, grade I vs. III

# refers to the progression of symptoms during hospitalization.

admission to surgery was 7 days (IQR: 5–9). Table 2 summarizes the management and outcome of the patients. When comparing early and delayed AV groups, several parameters showed a significant difference, i.e., duration under dialysis (three patients in the delayed AV group required dialysis), fluid infusion, antibiotics at admission, and length of hospital stay, while no significant difference was found in the other examined parameters (Table 3).

Table 4 summarizes the biological abnormalities recorded at admission and during hospitalization. Biological parameters at admission showed defibrinogenation in all cases,

**Table 3. Management and outcome of patients.**

| | | Total | | Early AV | | Delayed AV | p |
|---|---|---|---|---|---|---|---|
| **Parameter** | **Nb** | **Result** | **Nb** | **Result** | **Nb** | **Result** | |
| Dialysis | 119 | 3 (2.5%) | 45 | 0 (0%) | 74 | 3 (4.1%) | 0.289 |
| Duration under dialysis, days | 3 | 7 (6–55) | 0 | - | 3 | 7 (6–55) | - |
| Fluid infusion | 119 | 71 (59.7%) | 45 | 32 (71.1%) | 74 | 39 (52.7%) | 0.047 |
| Fluid infusion volume, ml | 71 | 1000 (1000–2000) | 32 | 1000 (1000–1500) | 39 | 1000 (1000–2000) | 0.633 |
| Antibiotics at admission | 119 | 42 (35.3%) | 45 | 10 (22.2%) | 74 | 32 (43.2%) | 0.020 |
| Time from SB to AV | 119 | 9:15 (5:32–17:47) | 45 | 5:00 (4:00–6:00) | 74 | 12:50 (9:16–23:00) | 0.000 |
| Adverse reaction to AV | 119 | 15 (12.6%) | 45 | 5 (11.1%) | 74 | 10 (13.5%) | 0.702 |
| Blood transfusion | 119 | 5 (4.2%) | 45 | 2 (4.4%) | 74 | 3 (4.1%) | 0.918 |
| Surgery | 119 | 27 (22.7%) | 45 | 9 (20%) | 74 | 18 (24.3%) | 0.585 |
| Time from admission to surgery, days | 27 | 7 (4–9) | 9 | 5 (4–6) | 18 | 7 (6–9) | 0.062 |
| Necrosectomy | 27 | 12 (44.4%) | 9 | 3 (33.3%) | 18 | 9 (50%) | 0.411 |
| Infection | 119 | 31 (26.1%) | 45 | 9 (20%) | 74 | 22 (29.7%) | 0.241 |
| Length of hospital stay, days | 119 | 7 (5–11) | 45 | 6 (4–10) | 74 | 8 (5–13) | 0.038 |

Nb: the number of cases in whom the parameter was analyzed, Values are expressed as number and percentages or median and interquartile range

thrombocytopenia in 31 cases (26.1%), hemolysis in 44 cases (37%), and rhabdomyolysis in 15 cases (12.6%). International Normalized Ratio was >2 in 105 cases (88.2%), and activated partial thromboplastin time (aPTT) was >1.5 in 71 cases (59.7%). When comparing the two groups, i.e., early (<6h) and delayed (>6h) AV administration, time from SB to the normalization of several clotting parameters (fibrinogen, INR, aPTT, factor II, and factor V) was shorter in patients receiving AV within the first six hours after the SB (Table 3). In contrast, no difference was observed regarding recovery of these parameters when comparing the time lapse between AV administration and recovery (Table 3). The rate of patients with a detectable fibrinogen dosage (>0.35 g/L) at admission was higher in the late AV group (35.6% *vs.* 15.6%; p = 0.018). Fig 3 shows the time from SB and AV administration to the normal value of fibrinogen, Factor II, and Factor V according to the time of AV initiation (≤6h *vs.* >6h). Fig 4 shows the time from SB and AV administration to the normal value of fibrinogen according to the delay in AV administration (≤6h *vs.* >6h). As a general trend, the time to correct alterations in some clotting factors and INR after the SB is more prolonged in the delayed AV group, although when the analysis was done from the time of AV administration, no significant difference was observed between the groups. Thus, once AV is administered, clotting alterations recover at a similar rate between the groups.

## Discussion

Our study shows that patients from rural areas in French Guiana took a longer time to receive AV after SB as compared to patients suffering the bite in locations close to Cayenne. Also, it shows that patients receiving AV within the first 6h after the SB were more likely to have a shorter time to normalize clotting parameters. However, the time between AV administration and recovery of clotting parameters was similar for the two groups.

Snakebite envenoming is an acute emergency requiring an early care delivery [7], including the early use of AV, ideally within the first six hours of envenoming [8–10]. In this context, our study emphasizes the vast difference in time to receive AV in French Guiana according to the geographic region where the bite occurred. Patients in remote zones must attend the closest local healthcare center before being evacuated to Cayenne to receive the AV. However, the

**Table 4. Biologic abnormalities recorded at admission and during hospitalization.**

| Parameter | Total Nb | Total Result | Early AV Nb | Early AV Result | Delayed AV Nb | Delayed AV Result | p |
|---|---|---|---|---|---|---|---|
| Rhabdomyolysis | 119 | 15 (12.6%) | 45 | 3 (6.7%) | 74 | 12 (16.2%) | 0.161 |
| Hemolysis | 119 | 44 (37%) | 45 | 16 (35.6%) | 74 | 28 (37.8%) | 0.803 |
| Time from SB to resolved hemolysis | 34 | 24:44 (20:15–37:07) | 13 | 21:00 (16:20–25:24) | 21 | 27:00 (24:00–46:00) | 0.008 |
| Presence of schistocytes | 31 | 3 (9.7%) | 13 | 1 (7.7%) | 18 | 2 (11.1%) | 1.000 |
| Defibrinogenation | 119 | 119 (100%) | 45 | 45 (100%) | 74 | 74 (100%) | - |
| Time from SB to normal fibrinogen | 112 | 26:55 (22:00–36:02) | 45 | 23:27 (20:00–27:10) | 67 | 31:23 (24:00–45:05) | 0.000 |
| Time from AV to normal fibrinogen | 112 | 15:52 (11:58–21:55) | 45 | 18:00 (14:26–21:54) | 67 | 14:10 (10:16–22:10) | 0.069 |
| Fibrinogen dosage>0.35 g/L on admission | 118 | 33 (28%) | 45 | 7 (15.6%) | 73 | 26 (35.6%) | 0.018 |
| INR>2 | 119 | 105 (88.2%) | 45 | 43 (95.6%) | 74 | 62 (83.8%) | 0.053 |
| Time from SB to normal INR | 98 | 23:17 (17:31–34:37) | 41 | 18:30 (16:14–30:30) | 57 | 25:30 (19:51–41:00) | 0.006 |
| Time from AV to normal INR | 98 | 12:00 (09:17–17:24) | 41 | 12:10 (11:14–17:07) | 57 | 11:58 (07:00–17:30) | 0.137 |
| aPTT>1.5 | 119 | 71 (59.7%) | 45 | 34 (75.6%) | 74 | 37 (50%) | 0.006 |
| Time from SB to normal aPTT | 70 | 17:24 (13:26–24:00) | 33 | 13:45 (11:00–17:18) | 37 | 21:37 (16:30–29:20) | 0,000 |
| Time from AV to normal aPTT | 70 | 08:45 (05:31–11:55) | 33 | 09:21 (05:40–11:43) | 37 | 08:10 (05:10–12:20) | 0.888 |
| Thrombocytopenia | 119 | 31 (26.1%) | 45 | 11 (24.4%) | 74 | 20 (27%) | 0.756 |
| Time from SB to normal platelet count | 24 | 66:30 (31:37–122:51) | 10 | 55:45 (34:15–115:08) | 14 | 68:05 (30:32–112:02) | 0.861 |
| Time from AV to normal platelet count | 24 | 57:34 (17:50–111:51) | 10 | 47:00 (19:57–110:32) | 14 | 57:34 (12:22–96:45) | 0.558 |
| Abnormal Factor II | 107 | 53 (49.5%) | 45 | 23 (51.1%) | 62 | 30 (48.4%) | 0.781 |
| Time from SB to normal FII | 29 | 32:35 (17:40–41:50) | 13 | 20:15 (16:10–27:00) | 16 | 37:35 (32:28–48:45) | 0.010 |
| Time from AV to normal FII | 29 | 17:15 (09:30–25:10) | 13 | 14:40 (09:30–20:13) | 16 | 20:45 (13:15–25:50) | 0.211 |
| Abnormal Factor V | 108 | 66 (61.1%) | 45 | 33 (73.3%) | 63 | 33 (52.4%) | 0.028 |
| Time from SB to normal FV | 47 | 24:50 (17:50–35:10) | 24 | 18:28 (15:20–25:15) | 23 | 31:30 (24:45–38:20) | 0.000 |
| Time from AV to normal FV | 47 | 14:00 (09:55–21:00) | 24 | 12:05 (09:19–16:15) | 23 | 17:30 (11:25–22:30) | 0.154 |
| Abnormal Factor VII | 107 | 36 (33.6%) | 45 | 13 (28.9%) | 62 | 23 (37.1%) | 0.375 |
| Time from SB to normal FVII | 13 | 28:00 (18:00–44:50) | 4 | 41:15 (22:59–62:25) | 9 | 28:00 (18:00–41:50) | 0.643 |
| Time from AV to normal FVII | 13 | 15:20 (08:30–26:50) | 4 | 30:43 (08:27–54:10) | 9 | 15:20 (08:30–16:30) | 0.355 |
| Abnormal Factor X | 104 | 27 (26%) | 44 | 11 (25%) | 60 | 16 (26.7%) | 0.848 |
| Time from SB to normal FX | 11 | 32:10 (18:30–35:10) | 4 | 21:28 (11:49–33:35) | 7 | 33:40 (26:05–35:10) | 0.345 |
| Time from AV to normal FX | 11 | 19:30 (09:13–24:20) | 4 | 16:58 (08:19–27:46) | 7 | 19:30 (10:03–22:25) | 0.850 |

Nb: the number of cases in whom the parameter was analyzed, Values are expressed as number and percentages or median and interquartile range, SB: snakebite, AV: antivenom, INR: International Normalized Ratio, aPTT: activated partial thromboplastin time

time for medical evacuation of patients is variable, since it depends on the helicopter availability and weather conditions. This disparity in time from SB to AV allowed us to compare an early AV group (≤6h from SB) and a delayed AV group (>6h). In similar contexts, most studies showed higher effectiveness of AV when administered early after the SB with significant improvement in clinical and biological parameters and better outcomes [11,18,19]. Thus, reducing the delay in treatment should be principally based on providing remote healthcare centers with AV. The critical point is that AV must go to patients, instead of patients having to travel long distances to receive antivenom. In French Guiana, two recent studies investigated the effectiveness of Antivipmyn Tri in the treatment of SB [2,6]. Heckmann et al. reported a median time from the SB to AV of 11h (IQR: 6–20), and only 11 patients received the AV before the sixth hour [6]. In another study, Resiere et al. found that the median time from the SB to AV was 9h (IQR: 5–21) [2].

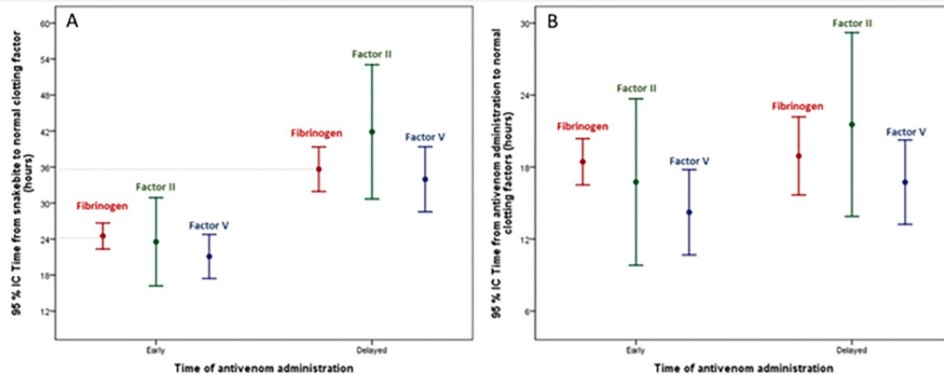

**Fig 3.** Time from SB (Plot A) and antivenom administration (Plot B) to the normal value of fibrinogen, Factor II, and Factor V according to the time between SB and antivenom administration (≤6h *vs.*>6h). The number of cases in whom the parameter was analyzed (Total, Early/Delayed) was: factor II (29, 13/16), factor V (47, 24/23), fibrinogen (112, 45/67). (Values are presented as mean and 95%CI).

French Guiana is a 83.534 km$^2$ department with an estimated population of 283,540 inhabitants. On average, 80% of the population lives in the coastal zone, and in suburban areas [20]. The remaining 20% live in remote areas where the huge disparities in healthcare availability and access to the nearest health care facility might be counted in hours or (sometimes) in days. Indeed, the one-trip access time to the nearest hospital can be about 10 to 20 min in the urban locations (by car), and it can go up to 100 min in isolated zones (by helicopter) [21]. In our study, there were two groups, i.e., patients from urban and suburban sites, and those from remote areas. In the urban and suburban groups, patients came directly to the ED and benefited from an early AV treatment (median time <3h). In the remote sites group, patients attended to the local healthcare center before being transferred to Cayenne hospital. Sometimes, patients must go through the forest or rivers to reach the healthcare center extending the delay in hospitalization and AV therapy (median time up to 19h). Consequently, the availability of AV in remote health centers is essential to shorten the delay to treatment. Once AV is administered in these remote health facilities, patients can be transferred to the hospital for further monitoring and care. These differences in time to attend a health facility and receive appropriate treatment are typical examples of inequitable access to care in the Guianese population. Similar reports from the Brazilian Amazon region were published, describing the

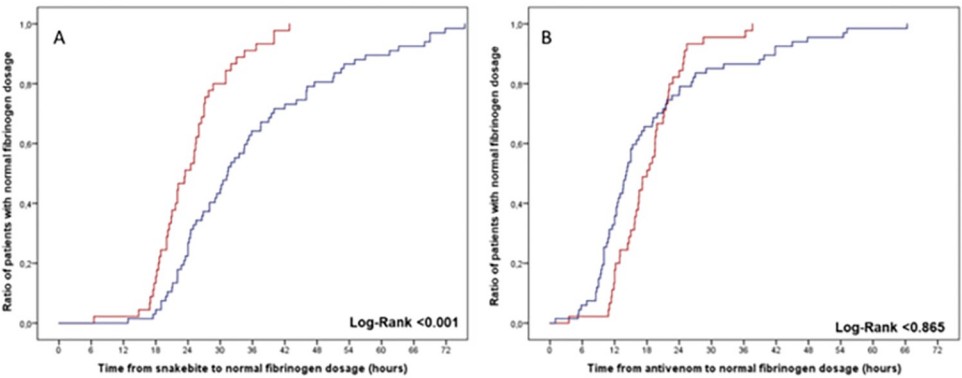

**Fig 4.** Time from SB (Plot A) and antivenom administration (Plot B) to the normal value of fibrinogen concentration according to the time between SB and antivenom administration [≤6h (red line, n = 45) *vs.* >6h (blue line, n = 67)].

limited access to health facilities and to antivenoms in the remote rural populations and rising the need for public health interventions [19,22,23]. In contrast, in a recent study in Costa Rica, Montoya-Vargas et al. [24] interviewed 96 pharmacists from 55 different healthcare facilities to investigate the way AV are managed by the public health system. Overall, participants reported AV availability at all levels of care and patients take less than 3h to medical assistance in the majority of cases. In FG, the AV is administered under a "compassionate" authorization for use by the French Agency for Drug Safety (Agence Française de Sécurité des Médicaments). This system allows the use of drugs without marketing authorization in France under specific conditions. It prohibits stocking the drug in remote health care centers and including patients in clinical trials. Currently, we are working together with the hospital pharmacists, the French Guiana Health Authority, and the French Agency for Drug Safety to get derogatory advice to provide remote health care centers with AV. European and French laws require that AV be administered by a nurse supervised by a doctor who can manage adverse reactions including anaphylactic shock. Thus, some French Guiana HCCs are planned to be provided by emergency doctors and trained nurses for AV administration and urgent snake-bitten patients' management.

AV should be administered in a health facility where acute adverse reactions can be treated. For this, before introducing AV in remote areas, some points have to be resolved. The first is to establish a management procedure for AV supplying and storage in remote areas to avoid disruption. In the case of liquid antivenoms, the maintenance of the cold chain must be ensured. Second, the validation of a protocol for AV use, including the method to grade the severity of envenomings, the prerequisites for AV administration, the duration of the monitoring before and after the immunotherapy, the modalities of preparation and injection of the AV, and the protocol to follow in the event of adverse effects. The criteria for evacuation to a hospital setting and the degree of emergency concerning the transfer time should be carefully considered. Indeed, non-predictable adverse effects can be observed during AV administration requiring close monitoring during the therapy and over [2,6]. Overall, an optimal approach to manage SB envenoming in FG must include antivenom availability and trained staff in remote healthcare facilities [20]. Furthermore, hospitals and healthcare authorities should promote comparative studies on the safety and efficacy of different AVs available in the region.

In agreement with previous studies in French Guiana, *B. atrox* was responsible for the majority of bites when the culprit snake was identified [2]. Likewise, the main clinical manifestations observed in patients included in this study were similar to those described in earlier works for *B. atrox*, i.e., local edema and pain, local hemorrhage, blisters, systemic bleeding and acute kidney injury [2,25]. The frequency of early adverse reactions to AV administration was low (12.6%), in agreement with previous studies in FG using this antivenom [2,6].

In our study, we compared the evolution of patients receiving AV within the first 6h after the bite with those receiving it at later time intervals, with emphasis on coagulopathy. The time from SB to correct coagulation disorders was more prolonged in the delayed AV group, thus highlighting the higher risk of these patients to suffer hemorrhage and other systemic complications. Previous studies in FG evaluated the restoration of clotting parameters comparing patients who received this AV and those who did not. Heckman et al. [6] described that the time from SB to reach fibrinogen concentration of 100 mg/dL was similar for the two groups (22 and 24h), although the fibrinogen recovery was faster beyond 30h in the AV group. In contrast, Resiere et al. [2] reported significant differences, since patients receiving AV reached 100 mg/dL fibrinogen concentration 25:30 h after SB, while this time was 47:00 h in those who did not receive AV.

On the other hand, when the time lapse from AV administration to recovery of clotting parameters was analyzed in our study, no significant difference was observed between the

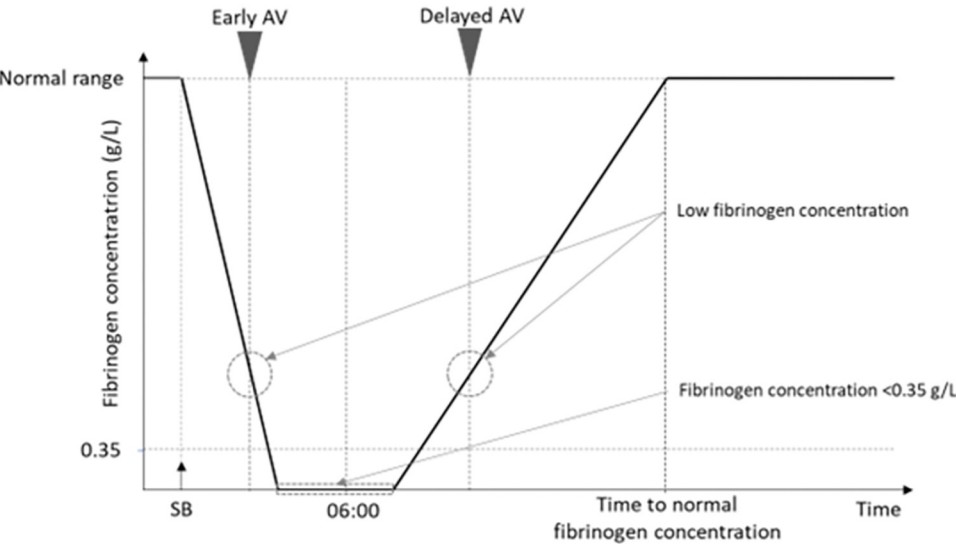

**Fig 5. Scheme of the proposed fibrinogen concentration kinetics following SB by *B. atrox*.** Venom induces a consumption coagulopathy, associated with a drastic drop in plasma fibrinogen concentration. After a time lapse, fibrinogen concentration starts to increase as a result of synthesis by the liver. Once AV is administered, venom toxins are neutralized and fibrinogen levels continue to increase until reaching normal concentration.

groups. Thus, once AV is administered, recovery from coagulation disturbances had an overall similar time course in the two groups. The time lapse to reach normal fibrinogen concentration after AV administration in our study was 15:52 h (11:58–21:55). The time of recovery from hypofibrinogenemia after AV administration in envenomings by *B. atrox* in Brazil and *Bothrops asper* in Colombia was estimated at 48h and 12-24h, respectively [25,26]. Thrombocytopenia is another common manifestation of *B. atrox* envenomings, and was observed in 26.1% of the patients included in this study. Interestingly, no significant differences were observed between the two groups in the time from SB or from AV administration required to recover normal platelet counts. It is necessary to further explore, at the preclinical and clinical levels, the ability of the current and other AVs to neutralize this relevant effect.

Following SB by *B. atrox*, the plasma fibrinogen concentration rapidly drops to very low levels due to venom-induced consumption coagulopathy. Two simultaneous processes determine the recovery of fibrinogen levels, i.e., the neutralization of venom procoagulant enzymes by AV and the replenishment of fibrinogen by the liver. Thus, it is hypothesized that in the delayed AV group there has been an increased synthesis of fibrinogen before AV administration (Fig 5). This hypothesis is supported by the finding that the percentage of patients with a detectable fibrinogen level at admission was higher in the late AV group than in the early AV group. However, despite the ongoing synthesis of fibrinogen by the liver, the neutralization of procoagulant toxins by AV is required to speed the recovery of coagulation parameters, as evidenced in previous studies with viperid SB envenomings in which patients receiving and not receiving AV were compared [2,27,28].

Our study has several limitations. First, it is an observational study with a small sample. Second, it includes patients with grade I envenoming (46.4%), limiting the evaluation of time to AV on systemic effects and skin necrosis and blisters. Indeed, in some not frequent outcomes, sample size may not be sufficient to show statistical significance in the comparison between groups. Third, snake identification was made in only 47.3% of cases. Identification was based on the patient's description, photographs, or on physical examination of the captured snake.

In this context, we recommend against bringing and/or killing the snake in order not to increase the risk of new accidents. Additionally, the genus Bothrops, mainly *B. atrox* is reported to be the principal etiological agent of SB envenoming causing the vast majority of bites and fatalities in the Amazon region [1,25].

## Conclusion

Our results show that there is a delay in AV administration in many SB cases in FG. Such delay extends the time lapse in which patients are coagulopathic, hence increasing the risk of bleeding and related clinical complications. Accordingly, the time lost to administer AV is proportional to the time needed to normalize coagulation parameters. Once AV is administered, the time lapse to recover clotting parameters is similar regardless of the delay in the onset of AV infusion. Supplying remote healthcare facilities with AV with medical teams trained on its use and on the management of related adverse effects should be planned for better and optimal SB-envenoming care in FG.

## Supporting information

**S1 Data. Is: Snakebites database registered in Cayenne Hospital.**
(XLSX)

## Author Contributions

**Conceptualization:** Jean Marc Pujo, Flaubert NkontCho, Magalie Pierre-Demar, José María Gutiérrez, Dabor Resiere, Didier Hommel, Hatem Kallel.

**Data curation:** Stéphanie Houcke, Segolene Vauquelin, Guy Roger Lontsi Ngoula, Severine Matheus, Hatem Kallel.

**Formal analysis:** Didier Hommel, Hatem Kallel.

**Methodology:** Stéphanie Houcke, Jean Marc Pujo, Dabor Resiere, Didier Hommel, Hatem Kallel.

**Software:** Hatem Kallel.

**Supervision:** Didier Hommel, Hatem Kallel.

**Validation:** Flaubert NkontCho, Magalie Pierre-Demar, José María Gutiérrez, Dabor Resiere, Didier Hommel, Hatem Kallel.

**Visualization:** Didier Hommel, Hatem Kallel.

**Writing – original draft:** Stéphanie Houcke, Jean Marc Pujo, Segolene Vauquelin, Severine Matheus, Didier Hommel, Hatem Kallel.

**Writing – review & editing:** José María Gutiérrez, Dabor Resiere, Hatem Kallel.

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
