## [Decision Letter · Decision Letter 0]

16 Jan 2023

Dear Dr Kallel,

Thank you very much for submitting your manuscript "Effect of the time to antivenom administration on recovery from snakebite envenoming-related coagulopathy in French Guiana" for consideration at PLOS Neglected Tropical Diseases. As with all papers reviewed by the journal, your manuscript was reviewed by members of the editorial board and by several independent reviewers. In light of the reviews (below this email), we would like to invite the resubmission of a significantly-revised version that takes into account the reviewers' comments. 

We cannot make any decision about publication until we have seen the revised manuscript and your response to the reviewers' comments. Your revised manuscript is also likely to be sent to reviewers for further evaluation.

Sincerely,

Wuelton M. Monteiro, Ph.D.

Section Editor

Wuelton Monteiro

Section Editor

Reviewer's Responses to Questions

**Key Review Criteria Required for Acceptance?**

**Methods**

-Are the objectives of the study clearly articulated with a clear testable hypothesis stated?

-Is the study design appropriate to address the stated objectives?

-Is the population clearly described and appropriate for the hypothesis being tested?

-Is the sample size sufficient to ensure adequate power to address the hypothesis being tested?

-Were correct statistical analysis used to support conclusions?

-Are there concerns about ethical or regulatory requirements being met?

Reviewer #1: Methods:

Add more info regarding Antivipmyn Tri®. Type of antivenom (Fab?) and the snake venoms that it neutralizes. Add info also regarding number of recommended vials. Besides that, the methods are very detailed.

Reviewer #2: The study proposed an analysis of accidents that occurred in French Guiana with a difference between early and delayed 6-hour VA administration. This context is interesting since 6 hours is a gold standard for receiving VA and after this time the risk of complications gradually increases.

Regarding the classification of poisoning, the authors could describe how it is done according to the conclusions of the 131 international symposium held in French Guiana in 2017 cited in the method to make it clearer.

Given the local epidemiology, how was the species of snake involved in the accident defined? The antivenom used neutralizes venom of which genus/species? How many vials are used for each accident classification?

What is the maximum follow-up time for patients?

What were the patient exclusion criteria set at baseline?

What was the estimated number of cases in the period evaluated? Did the authors make a sample calculation?

Did the laboratory tests have a frequency and interval period to be performed?

Reviewer #3: This is an interesting study about the effect of the time to antivenom (AV) administration on recovery of the effects of snakebite envenoming in patients receiving AV early (in the first 6 hours after bite) or lately (6 hours or more after bite).

Objectives are clearly stated, and the study design is in accordance with the purposes. It is not described how the sample size was defined. It may be inferred that number of patients was obtained from the period outlined in the study. However, for some not frequent outcomes, sample size may not have been sufficient to show statistical significance in the comparison between groups, such as necrosis, blisters and dialysis. Authors indicate this limitation, suggesting the possibility of confounding factors, but do not explain which ones. Readers would benefit from greater detail in this respect.

**Results**

-Does the analysis presented match the analysis plan?

-Are the results clearly and completely presented?

-Are the figures (Tables, Images) of sufficient quality for clarity?

Reviewer #1: The results are fine.

Reviewer #2: In line 206 the authors described that 52 patients had the snake identified, was it by photo or did they bring the snake? 51 snakes were Bothrops atrox and what was the other 1 species?

Line 211 Do the authors describe the main signs and symptoms of the patients, is this information from the patient's arrival at the hospital or at another time? The clinical information is in general, wouldn't it be better to divide into the group of early AV and delayed AV as in the rest of the results?

Table 1 has the definition of GRADE I, II and III, please describe clearly in the methodology to improve understanding of this classification.

In table 3, not all patients underwent all laboratory tests, some of these, for example Factor X: out of 104, only 11 were evaluated in the table for Time from SB to normal and Time from AV to normal FX. Why was information not presented on the 104 patients who had this abnormal value? Thus, the small number of results would not be good for conclusions and analyzes due to possible biases. Is this the correct interpretation or am I mistaken?

The fact of not having the monitoring of the abnormal exams of the 119 patients left me confused in the construction of figures 3 and 4, how many patients were considered to perform the graph calculations?

Reviewer #3: Statistical tests were correctly applied, but some numerical variables, such as fibrinogen and platelets, were not analyzed according to their medians. Authors preferred to convert them in discrete variables (number and percentage of patients with defibrinogenation, number and percentage of patients with thrombocytopenia). Would the comparison be different if variables se variables present different 

Rhabdomyolysis and hemolysis were included as variables to be studied, but their meaning in Bothrops envenoming in FG was not discussed. Laboratory tests were performed but values of CK (for rhabdomyolysis), LDH, bilirubin and haptologlobin (for hemolysis) were omitted. May these parameters did not show difference between groups, but authors should give some explanation.

**Conclusions**

-Are the conclusions supported by the data presented?

-Are the limitations of analysis clearly described?

-Do the authors discuss how these data can be helpful to advance our understanding of the topic under study?

-Is public health relevance addressed?

Reviewer #1: Yes, the conclusion support the study.

Reviewer #2: Some parameters were associated in the statistical analysis, such as the first visit in a remote area, snake identification and local hemorrhage. However, only remote area care was well discussed, could the authors include the other points mentioned in the discussion as well?

As for Factor V, aPTT, Defibrinogenation, I need clarification on the number of patients who completed the exams to issue a more correct opinion, as I was in doubt in interpreting the information in Table 3 and graphs 3 and 4.

Figure 5 and the explanation given in the text were very good, however, this would not be the appropriate place for this information. So I suggest that this be in the methodology section.

In the limitation of the study, the authors describe that the entry of GRADE I patients that limited the evaluation of the time to the effects of VA, could the authors how this limited the study? Were the patients discharged early, not allowing the collection of exams and clinical evaluation to monitor the complete recovery of parameters, or was there another reason? Regarding the limitation of the identification of the snake in 47.3% of the cases - in line 207 43.7% were described - the authors could have a moment in the discussion that the stimulus of bringing and/or killing the snake should not be done in order not to increase the risk of new accidents and that the classification of the type of snake responsible for the envenomation is carried out by the patient's clinic, that is, this should not be considered a limitation of the study.

Reviewer #3: It is interesting the theory proposed to explain the higher percentage of patients with fibrinogen levels > 0.35 g/L in those who received AV late, compared to those treated early. Despite the correct recommendation for AV administration in any circumstance of systemic envenoming, could grade I patients suggest spontaneous recovery of coagulopathy in patients mildly envenomed? 

The availability of antivenom in remote health facilities is strongly emphasized as a key aspect to reduce morbidity and mortality of snakebite envenoming in rural and traditional populations. This is an important contribution of study and could be complemented with some discussion regarding the type of health professional to be responsible for the administration of antivenom. Should be a medical doctor? In places where this type of professional is not available, could a nurse of health agent administer AV? What about the risk of complications of an adverse reaction to antivenom if used in a primary care unit?

**Editorial and Data Presentation Modifications?**

Reviewer #1: Minor revision

Reviewer #2: The authors need to clarify the laboratory results.

Reviewer #3: Line 139: "medical observation" is more appropriate than "surveillance"

Table 1: correct typing "snake identification"

Table 3: include legend, similarly to table 1 and 2.

**Summary and General Comments**

Reviewer #1: The article intitled Effect of the time to antivenom administration on recovery from snakebite envenoming- related coagulopathy in French Guiana is interesting. Although not novel (other studies have investigated the same) there is a lack of literature info regarding the snakebite treatment FG. The manuscript is easy to read; however, there are some grammar and other mistakes. I encourage authors read it again and have the article read by a native speaker as well. I really liked Figure 1, it is very explicative. For me the main results are in Figure 3. Is the group the pioneer to show that? Please check the literature and make it clear. Based on that, I also encourage a quick discussion regarding the hospital costs comparing the time the patient stay in the hospital (receiving early and late AV).

Minor issues:

Line 38, 43, 107, 208,etc: change hr to h (check all the manuscript)

Lines 39-40: to correct laboratory coagulation parameters? Maybe to return to normal levels of coagulation. Please rewrite.

Line 42: change sites to areas

Line 43: change it to The time

Line 58: remove comma before such

Line 66: a loss of chance???? Please rewrite

Line 194: medevac???? See 320 too. 

Line 315: put together snakebite

Methods:

Add more info regarding Antivipmyn Tri®. Type of antivenom (Fab?) and the snake venoms that it neutralizes. Add info also regarding number of recommended vials.

Reviewer #2: The article is interesting mainly because it is a follow-up of patients, however some information is not clear to adequately respond to the objective of the study and understand the evolution of the clinical condition of the patients as described in the recommendations above.

Reviewer #3: (No Response)

PLOS authors have the option to publish the peer review history of their article (what does this mean?). If published, this will include your full peer review and any attached files.

Reviewer #1: No

Reviewer #2: No

Reviewer #3: No
---

## [Editor Report · Decision Letter 1]

8 Feb 2023

Dear Dr Kallel,

Thank you very much for submitting your manuscript "Effect of the time to antivenom administration on recovery from snakebite envenoming-related coagulopathy in French Guiana" for consideration at PLOS Neglected Tropical Diseases. As with all papers reviewed by the journal, your manuscript was reviewed by members of the editorial board and by several independent reviewers. In light of the reviews (below this email), we would like to invite the resubmission of a significantly-revised version that takes into account the reviewers' comments. 

We cannot make any decision about publication until we have seen the revised manuscript and your response to the reviewers' comments. Your revised manuscript is also likely to be sent to reviewers for further evaluation.

Sincerely,

Wuelton M. Monteiro, Ph.D.

Section Editor

Wuelton Monteiro

Section Editor
---

## [Decision Letter · Decision Letter 2]

14 Mar 2023

Dear Dr Kallel,

We are pleased to inform you that your manuscript 'Effect of the time to antivenom administration on recovery from snakebite envenoming-related coagulopathy in French Guiana' has been provisionally accepted for publication in PLOS Neglected Tropical Diseases.

Best regards,

Wuelton M. Monteiro, Ph.D.

Section Editor

Wuelton Monteiro

Section Editor

Reviewer's Responses to Questions

**Key Review Criteria Required for Acceptance?**

**Methods**

-Are the objectives of the study clearly articulated with a clear testable hypothesis stated?

-Is the study design appropriate to address the stated objectives?

-Is the population clearly described and appropriate for the hypothesis being tested?

-Is the sample size sufficient to ensure adequate power to address the hypothesis being tested?

-Were correct statistical analysis used to support conclusions?

-Are there concerns about ethical or regulatory requirements being met?

Reviewer #2: The description of the sample and the evaluated patients was better described and is more coherent with the objectives and results.

**Results**

-Does the analysis presented match the analysis plan?

-Are the results clearly and completely presented?

-Are the figures (Tables, Images) of sufficient quality for clarity?

Reviewer #2: The authors added information and tables that clarified the information from the results.

**Conclusions**

-Are the conclusions supported by the data presented?

-Are the limitations of analysis clearly described?

-Do the authors discuss how these data can be helpful to advance our understanding of the topic under study?

-Is public health relevance addressed?

Reviewer #2: It's adequate.

**Editorial and Data Presentation Modifications?**

Reviewer #2: Accept

**Summary and General Comments**

Reviewer #2: The authors met most of the recommendations and the manuscript is significantly improved.

PLOS authors have the option to publish the peer review history of their article (what does this mean?). If published, this will include your full peer review and any attached files.

Reviewer #2: No

---

## [Editor Report · Acceptance letter]

20 Apr 2023

Dear Dr Kallel,

We are delighted to inform you that your manuscript, "Effect of the time to antivenom administration on recovery from snakebite envenoming-related coagulopathy in French Guiana," has been formally accepted for publication in PLOS Neglected Tropical Diseases.

Best regards,

Shaden Kamhawi

co-Editor-in-Chief

Paul Brindley

co-Editor-in-Chief
